# Demonstration of a superconducting diode-with-memory, operational at zero magnetic field with switchable nonreciprocity

Taras Golod [1] & Vladimir M. Krasnov [1]✉

Diode is one of the basic electronic components. It has a nonreciprocal current response, associated with a broken space/time reversal symmetry. Here we demonstrate prototypes of superconducting diodes operational at zero magnetic field. They are based on conventional niobium planar Josephson junctions, in which space/time symmetry is broken by a combination of self-field effect from nonuniform bias and stray fields from a trapped Abrikosov vortex. We demonstrate that nonreciprocity of critical current in such diodes can reach an order of magnitude and rectification efficiency can exceed 70%. Furthermore, we can easily change the diode polarity and switch nonreciprocity on/off by changing the bias configuration and by trapping/removing of a vortex. This facilitates a memory functionality. We argue that such a diode-with-memory can be used for a future generation of in-memory superconducting computers.

---

[1] Department of Physics, Stockholm University, AlbaNova University Center, SE-10691 Stockholm, Sweden. ✉email: Vladimir.Krasnov@fysik.su.se

Large computation facilities, such as big data centers and supercomputers have become major energy consumers with a power budget often in excess of 100 MW. It has been argued that a small fraction of this power would be sufficient for cooling down the facility to cryogenic temperatures, suitable for operation of superconductors (SC)[1]. SC electronics would not only enable effective utilization of energy by removing resistive losses, it could also greatly enhance the operation speed. Since there is no resistance, $R = 0$, the $RC$ time constant is no longer a limiting factor. The ultimate operation frequency is determined by the SC energy gap. For many SCs it is in the THz range[2]. This enables clock frequencies several orders of magnitude higher than for modern semiconducting electronics. Such perspectives has lead to a renewed interest in development of a digital SC computer[1,3–7].

Diode is one of the primary electronic components. Its non-reciprocal current–voltage (I–V) characteristics allow rectification of alternating currents, which is necessary for signal processing and ac–dc conversion. Diodes can be also used as building blocks for Boolean logics in digital computation. SC diodes should have strongly asymmetric critical currents, $|I_{c+}| \neq |I_{c-}|$. It is well known that nonreciprocity may appear in spatially asymmetric SC devices[8,9]. SC diodes, based on spatially nonuniform Josephson junctions (JJs), were demonstrated long time ago[10]. Also SC ratchets[11], rectifying motion of either Josephson[12–15] or Abrikosov[16–23] vortices, were intensively studied. However, such spatially asymmetric devices operate only at finite magnetic fields, while computer components should work at zero field. Non-reciprocity at $H = 0$ is prohibited by the time-reversal symmetry, which requires invariance of electromagnetic characteristics upon simultaneous flipping of current and magnetic field[10,24]. Therefore, zero-field SC diode requires breaking of both space and time-reversal symmetry.

Recently it was shown that nonreciprocity can be induced in noncentrosymmetric SC by spin–orbit interaction (SOI)[25–29]. This renewed search for diode effects in noncentrosymmetric SC[25,29–31] and heterostructures[32–34]. SOI can induce asymmetry of either resistance in the fluctuation region near $T_c$[25–27,29,31,35], or supercurrent at low $T$[32,33,36–40]. However, SOI-based diodes require significant magnetic field. In several works zero-field SC diode operation was reported[34,37], involving additional nontrivial effects. In this respect, nonreciprocity can be a tool for investigation of unconventional SC[35–40].

In this work, we demonstrate prototypes of SC diodes with a large and switchable nonreciprocity of supercurrent at zero magnetic field. They are made of a conventional Nb SC and contain cross-like planar Josephson junctions with additional electrodes and an artificial vortex trap. Nonreciprocity is induced by a combination of self-field effect from asymmetric bias and stray fields from trapped Abrikosov vortex (AV). We demonstrate that the ratio, $|I_{c+}/I_{c-}|$, of such diodes can reach an order of magnitude and rectification efficiency can exceed 70%. Furthermore, we can switch nonreciprocity on and off, as well as change diode polarity in one and the same device. This is achieved by trapping/removing either a vortex, or an antivortex, and/or by changing the bias configuration. This facilitates memory functionality. We argue that such a diode-with-memory can be used for a new generation of superconducting in-memory computers.

## Results

**The concept**. We consider the simplest case of a short JJ with the length $L < 4\lambda_J$, where $\lambda_J$ is the Josephson penetration depth. This allows neglecting of complex phenomena associated with screening effects and Josephson vortices[9,10,41]. Realization of zero-field SC diode requires breaking of space/time symmetry. Time-reversal leads to inversion of transport currents and magnetic fields generated by these currents. The role of an external field, $H$, is somewhat more tricky[24]. However, since it induces a spatial phase gradient in a JJ, it is connected with the spatial symmetry[41].

Our concept has two simple ingredients: (i) Utilization of a nonuniform bias for achieving nonreciprocity at finite fields[10]; and (ii) Shifting it to zero field by persistent stray fields from a trapped AV[41–43]. These effects are summarized in Fig. 1a and b. Here black lines represent the conventional Fraunhofer modulation of the critical current versus magnetic flux, $I_c(\Phi)$, for a uniform JJ without a vortex. In this case, there are both time-reversal, $I_{c+}(H) = |I_{c-}(H)|$, and space-reversal, $I_{c\pm}(H) = I_{c\pm}(-H)$, symmetries.

Nonuniformity of junction characteristics breaks the spatial symmetry. Most common are nonuniform critical current density, $J_c(x)$, and bias current, $I_b(x)$, distributions. Both lead to the appearance of self-field effect[8–10]: a nonuniformly distributed current generates magnetic field component parallel to $H$. Olive and blue lines in Fig. 1a represent calculated $I_c(\Phi)$ patterns (see Methods) for small and large self-field effects in a JJ with a

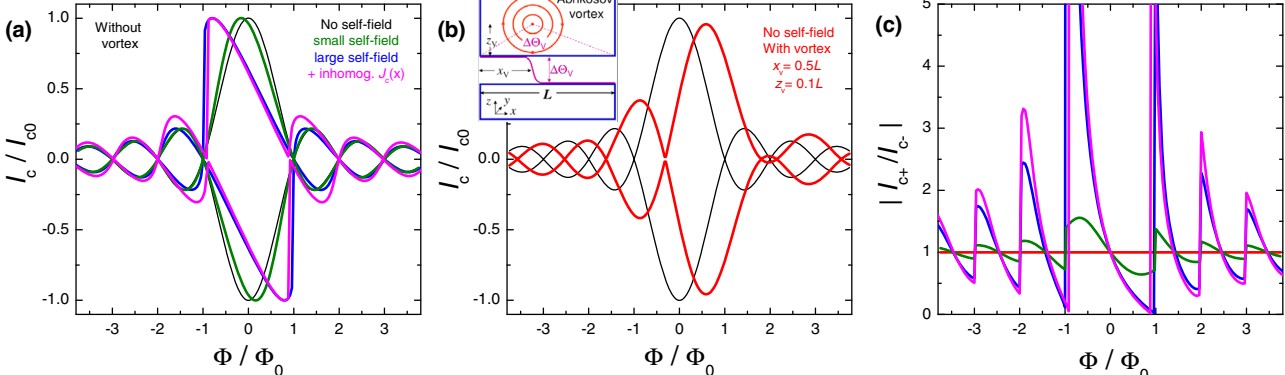

**Fig. 1 Numerical modeling of the two diode ingredients. a** Self-field phenomenon, induced by a spatial nonuniformity and (b) Flux offset by stray fields from a trapped Abrikosov vortex. **a** Simulated $I_c(\Phi)$ modulation: black—for a uniform JJ with constant critical current density, $J_c$, and bias, $I_b$; olive - for $J_c = const$ with a slightly nonuniform bias, $I_b(x)$, leading to appearance of self-field; blue—for $J_c = const$ with a strongly nonuniform bias, $I_b(x)$; magenta—the same as for blue with added V-shape spatial inhomogeneity of $J_c(x)$. **b** Simulated $I_c(\Phi)$ for a uniform JJ with a uniform bias without (black) and with a trapped antivortex (red) at $x_v = L/2$ and $z_v = 0.1L$. The inset represents a sketch of the vortex-junction configuration. **c** The nonreciprocity, $|I_{c+}(\Phi)/I_{c-}(\Phi)|$, for the cases from **a**, **b** in the same color palette. It is seen that nonuniformity induces nonreciprocity, but only at a finite field, $H \propto \Phi \neq 0$. Vortex stray fields offset and distort $I_c(\Phi)$, but do not induce nonreciprocity.

uniform critical current, $J_c(x)$ = const, but with a nonuniform bias, $I_b(x) \neq$ const. Magenta lines are calculated for the same $I_b(x) \neq$ const as for the blue curves, with additional nonuniformity of $J_c(x) \neq$ const. As discussed in ref. [10], asymmetric nonuniformities of both $J_c(x)$ and $I_b(x)$ tilt $I_c(H)$ patterns and lead to appearance of a nonreciprocity, $I_{c+}(H) \neq |I_{c-}(H)|$, at finite $H$. However, for any nonuniformity, the $I_c(H)$ modulation remains centrosymmetric, $I_{c+}(H) = -I_{c-}(-H)$. This is the consequence of space/time symmetry: simultaneous flipping of $I$ and $H$ is equivalent to looking at the same JJ from the back side and, therefore, should lead to the identical observation[10].

Red lines in Fig. 1b represent the $I_c(\Phi)$ modulation in a uniform junction with a trapped AV, placed symmetrically in the middle of the electrode, $x_v = L/2$, at a distance $z_v = 0.1L$ from the JJ (see the sketch in the inset). Stray fields from AV both distort and shift the $I_c(H)$ pattern[42–44]. This breaks the space-reversal, $I_{c\pm}(H) \neq I_{c\pm}(-H)$, but preserves the time-reversal, $I_{c+}(H) = |I_{c-}(H)|$, symmetry. Note that for short JJs this symmetry is preserved even for asymmetric vortex locations, $x_v \neq L/2$, but for long JJs all types of symmetries are removed due to appearance of Josephson vortices[41].

In Fig. 1c we plot the nonreciprocity, $|I_{c+}(\Phi)/I_{c-}(\Phi)|$, for the curves from panels (a) and (b). It is seen that uniform JJs without (black) or with (red) a vortex are reciprocal, $|I_{c+}/I_{c-}| = 1$. Nonuniform JJs exhibit the nonreciprocity, which grows with increasing the inhomogeneity of $J_c(x)$ and $I_b(x)$. Such Josephson diodes were studied earlier[10]. Their nonreciprocity could be very high: the largest peak at $\Phi/\Phi_0 \simeq -1$ for the magenta curve in Fig. 1c reaches two orders of magnitude. However, due to the centrosymmetric $I_c(H)$ there is no effect at $H = 0$.

For shifting the nonreciprocity to $H = 0$, we utilize persistent stray fields from a trapped AV. As shown in refs. [43,44], a flux offset introduced by AV in planar JJs is determined by the polar angle, $\Theta_v$, of the vortex within the junction (see the sketch in Fig. 1 b). It depends on the vortex location $(x_v, z_v)$, which is the tunning geometrical factor for the vortex-induced flux offset[43,44]. In Fig. 2 we show calculated variation of the $I_c(\Phi)$ modulation (left) and the nonreciprocity, $|I_{c+}(\Phi)/I_{c-}(\Phi)|$, (right panels) upon approaching AV towards the junction from (a) $z_v = \infty$ to (e) $z_v = 0.01L$ along the middle line, $x_v = L/2$. Simulations are done for the same nonuniform JJ depicted by the magenta line in Fig. 1. It is seen that with approaching the vortex to the JJ, the central nonreciprocal peak moves gradually from $\Phi/\Phi_0 \simeq -1$ toward 0 without significant reduction of the amplitude. At (d) $z_v = 0.05L$ it passes through $\Phi = 0$. This is the optimal geometrical configuration for a zero-field diode operation.

**Experimental verification.** Zero-field diodes are realized using four-terminal cross-like JJs. Figure 3a, b shows scanning electron microscope images of the two studied devices, D1 and D2. They have similar geometries. Each contains two planar JJs with $L = 5.6\ \mu m$, seen as horizontal lines, and a vortex trap—a hole with diameter ~ 50 nm, placed at $x_v \simeq L/2$ and $z_v \simeq 0.1L$ from JJ1. D1 is made from a Nb(70 nm)/CuNi(50 nm) bilayer with superparamagnetic CuNi, while D2 is made from a single Nb film (70 nm). Therefore, D1 contains proximity-coupled Nb-CuNi-Nb JJs and D2—variable thickness type constriction JJs, Nb-c-Nb. Both devices behave in a similar way, but Nb-c-Nb JJs have much larger $I_c R_n$, approaching 1 mV at low $T$[45]. This increases both the readout voltage and the upper operation frequency, $f_c = I_c R_n/\Phi_0$, which is advantageous for electronic applications. Details of junction fabrication, characterization and experimental setup can be found in Methods, Supplementary information and refs. [3,42,43,45–48]. Magnetic field is applied perpendicular to Nb film (in the $y$-direction).

As can be seen from Fig. 3b, the studied devices have a cross-like geometry with four electrodes (left, right, top, bottom). This allows a controllable introduction of a bias asymmetry[47]. In Fig. 3c we sketch three bias configurations for JJ1. The straight (bottom-to-top) bias does not generate $H_y$ field component, while biases over right/left corners do induce positive/negative self-fields, $H_y$, in the junction. This strongly affects junction characteristics and facilitates tunable introduction of spatial asymmetry.

Figure 3d, e shows $I_c(H)$ patterns of JJ1 in the vortex-free case, measured using the three bias configurations for (d) D1 and (e) D2. The straight bias (olive curves) leads to a regular Fraunhofer-type modulation. However, right (red) and left (blue) corner biases tilt $I_c(H)$ patterns in opposite directions due to the appearance of self-fields[10]. Figure 3f shows the $I–V$ characteristics at three fields indicated by dashed lines in Fig. 3d. At $H = 0$ (green) the $I–V$ is symmetric $I_{c+} = |I_{c-}|$, however, at $H \simeq \pm 0.8$ Oe, profound nonreciprocities appear, reaching an order of magnitude of either direction. This demonstrates that the cross-like geometry allows simple and controllable introduction of the spatial (bias) asymmetry and the associated nonreciprocity at $H \neq 0$.

Figure 4 summarizes diode performance for D1 with the right-corner bias (a) without AV, and (b) with a trapped vortex, or (c) antivortex. AV is controllably introduced and removed by short current pulses[3,44], as described in the Supplementary. Top panels show $I_c(H)$ modulations for JJ1 (black) and JJ2 (olive). Red lines represent numerical fits (see Methods). In Fig. 4a it is identical with the magenta curve in Fig. 1a. Fits in panels (b) and (c) are made for the actual geometry of the vortex trap and with AV-induced flux $\delta\Phi/\Phi_0 = \pm 0.47$ as a fitting parameter. It is consistent with $\Theta_{v1}/2\pi \simeq 0.44$.

Middle panels in Fig. 4 represent main experimental results of this work: the nonreciprocities $|I_{c+}/I_{c-}|$ and $I_{c-}/I_{c+}$ for both JJs on D1. It can be seen that without AV, $|I_{c-}/I_{c+}|$ at $H \simeq 0.8$ Oe exceeds an order of magnitude, while it is absent at $H = 0$. Introduction of AV shifts the maxima so that a significant nonreciprocity occurs at zero field, as indicated by cyan ovals in (b) and (c). The maximum for JJ2 in (b) exceeds a factor four. For JJ1 it is slightly offset (by ~0.1 Oe) but still exceeds a factor two at $H = 0$. In (c) nonreciprocity at $H = 0$ is more than three for both JJs. Note, that diode polarity is opposite for vortex, $I_{c+} > |I_{c-}|$, and antivortex, $I_{c+} < |I_{c-}|$. Moreover, in our cross-like devices, the polarity can also be flipped by changing bias configuration. In Fig. 4 we use the right-corner bias. If we change to the left-corner bias, both the self-field and polarity change the sign, as demonstrated in Fig. 3d, e. For the left-corner bias the polarity of vortex and antivortex states flips so that $|I_{c-}| > I_{c+}$ for the vortex and $I_{c+} > |I_{c-}|$ for the antivortex. All mentioned states are persistent and are achievable in one and the same device. Therefore, our diode is *switchable*. This enables a memory functionality[3] with three distinct states at $H = 0$: a reciprocal state "0" without AV, Fig. 4a, and states "+1" and "−1" with positive and negative polarities, shown in Fig. 4b, c. Such reconfigurability is a unique property of vortex-based devices[44].

Rectification is an important property of a diode. Bottom panels in Fig. 4 show rectified time-average dc-voltage, calculated for ac-bias with the amplitude $I_{ac} = 220\ \mu A$. Oscillatory field dependencies, with significant rectified voltages at central peaks, can be seen[10]. The maximum rectifiable voltage for the case when one side of the $I–V$ is fully open, $I_{ac} < I_c$, and the other is fully closed, $I_c = 0$, is $\langle V_{max}\rangle = I_{ac}R_n/\pi$. Central peaks for simulated red curves, which have nonreciprocities in the range of 30–50, are practically ideal. Experimental peaks for the vortex-free case (a) are exceeding 80% of that value for the two central peaks. The peaks at $H = 0$ in (b) and (c) exceed 70%, indicating good rectification efficiency of the diodes.

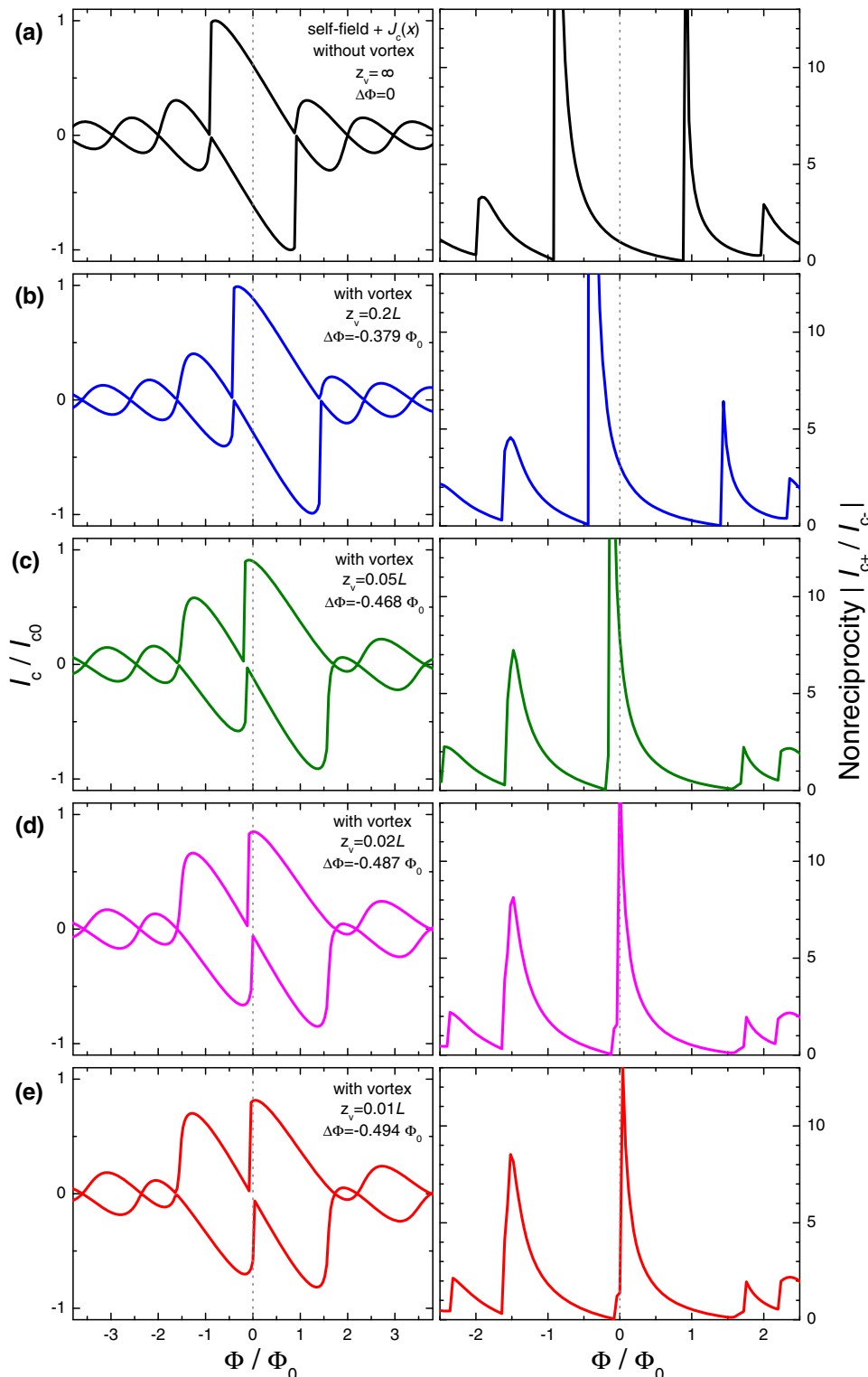

**Fig. 2 Numerical modeling of combined effects of nonuniformity and stray magnetic fields from a trapped vortex.** Left panels show simulated $I_c(H)$ patterns for a nonuniform JJ (the same as shown by magneta curves in Fig. 1) upon approaching an Abrikosov vortex to the JJ along the middle line, $x_v = L/2$, from **a** $z_v = \infty$ to **e** $z_v = 0.01L$. Right panels represent corresponding nonreciprocities, $|I_{c+}(H)/I_{c-}(H)|$. It is seen that upon approaching the vortex to the junction, growing stray fields progressively shifts nonreciprocal peaks. At a certain distance (**d**) the main peak passes through $H = 0$. This is the optimal geometrical configuration for a zero-field diode.

Figure 5 illustrates the ac-bias dependencies of rectification for D1 (a,b) and D2 (c,d). For clarity, we consider states with the maximum nonreciprocity at finite $H$. The top panel in Fig. 5a shows the $I–V$ (royal) of JJ1 on D1 with near maximum $|I_{c+}/I_{c-}|$. The bottom panel shows measured time dependencies of voltage,

for different ac-bias amplitudes. It is seen that for $|I_{c-}| < I_{ac} < I_{c+}$ only negative voltage appears during the ac-oscillation period. This leads to appearance of a negative time-average dc-voltage, $\langle V \rangle < 0$. The top panel in Fig. 5b shows the bias dependence of rectified voltage. It appears at $I_{ac} > |I_{c-}|$, grows linearly up to

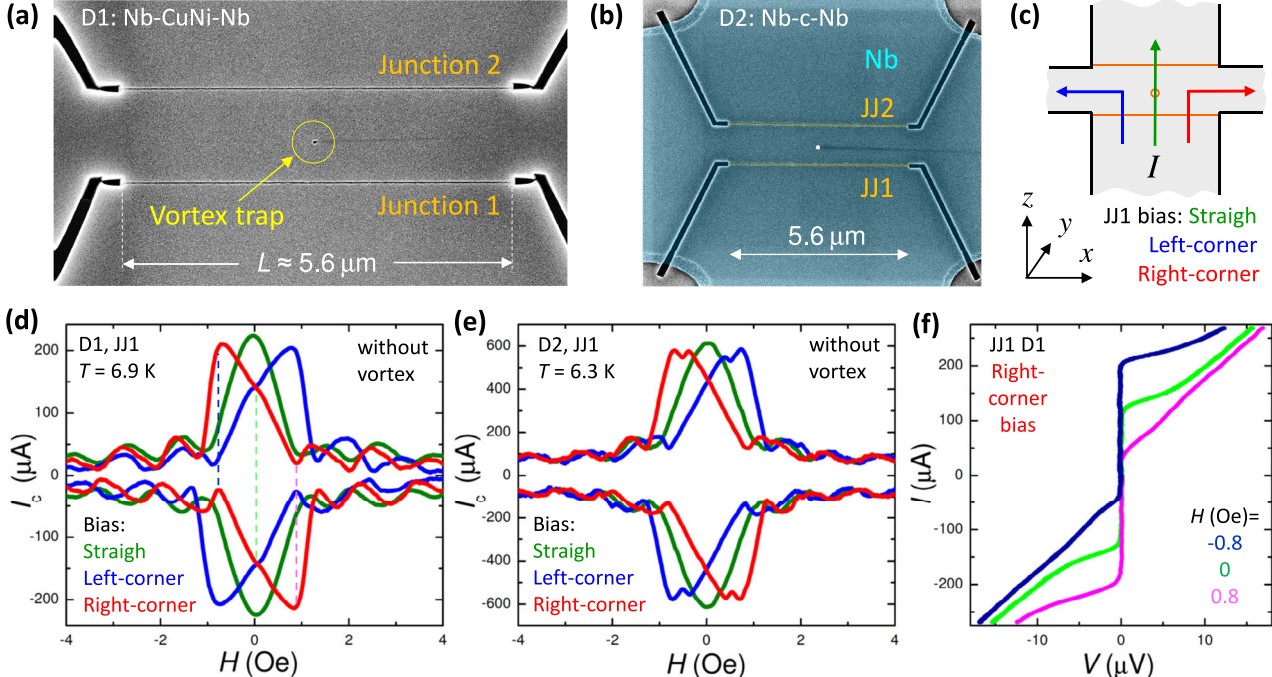

**Fig. 3 Device characterisation in the vortex-free case.** SEM images of the two studied devices: **a** D1 with Nb-CuNi-Nb junctions and **b** D2 with constriction-like Nb-c-Nb variable thickness bridges (false color). They have similar geometries and contain two planar junctions and a vortex trap. As seen from **b**, devices have cross-like geometry. This allows controllable variation of the self-field effect by changing bias configurations, as sketched in **c**. Field is applied perpendicular to the film, in the $y$-direction, $(x, y, z)$ is the right-handed coordinate system. Measured $I_c(H)$ patterns for JJ1 on **d** D1 and **e** D2 for three bias configurations. The straight bias (olive lines) does not induce self-field. Left (blue) and right (red) corner biases induce self-fields of opposite signs, causing tilting of the $I_c(H)$ patterns in opposite directions. **f** The $I–V$ curves for JJ1 on D1 at three magnetic fields marked by dashed lines in **d**. A profound nonreciprocity with a factor ~10 difference between $I_{c+}$ and $|I_{c-}|$ can be seen.

$I_{ac} = I_{c+}$ and then decreases due to progressive increase of positive voltages during the oscillation period, as seen from magenta and black $V(t)$ curves in Fig. 5a. The green line represents the ideal case with infinite nonreciprocity, $\langle V_{max} \rangle = I_{ac}R_n/\pi$. The bottom panel in Fig. 5b shows rectification efficiency with respect to the ideal case, $\langle V \rangle / \langle V_{max} \rangle$. It is seen that the maximum efficiency, achieved at $I_{ac} = I_{c+}$, exceeds 80%. The maximum rectification efficiency is close to $1 - 1/v$, where $v$ is the nonreciprocity of $I_c$.

Figure 5c, d demonstrates similar data for D2. Here we analyze the state, represented by the red $I–V$ with a maximum nonreciprocity of $|I_{c-}/I_{c+}|$. It has the opposite diode polarity, compared to Fig. 5a, c, resulting in $\langle V \rangle > 0$. The overall performance is similar to D1, except for the larger $I_cR_n$ of Nb-c-Nb JJs, which leads to proportionally larger rectified voltages and the upper-frequency range, $\sim I_cR_n/\Phi_0$ (see Supplementary information for additional clarifications).

## Discussion

We demonstrated operation of Josephson diodes-with-memory with large and switchable nonreciprocity at zero magnetic field. Our concept is based on utilization of nonuniform bias for inducing nonreciprocity and stray fields of Abrikosov vortex, trapped at a proper position, for shifting the nonreciprocity to zero field. It is shown that such diodes have very good performance. The measured nonreciprocity of critical current exceeds a factor 4 at zero field and is more that an order of magnitude at finite field. Numerical modeling indicates that these values can be improved by another order of magnitude by a careful design. The rectification efficiency exceeds 70% at zero field. This is good enough for realization of more complex logical Boolean devices, needed for a digital superconducting computer. It has already been demonstrated[3] that a very simple

geometry of such devices, which do not utilize SQUIDs, along with a nano-scale vortex size allows drastic miniaturization down to sub-micron dimensions. However, the most unique feature of our diodes is their switchability and tunability: (i) the nonreciprocity at $H = 0$ can be easily introduced/removed by trapping/removing Abrikosov vortices using short current pulses and (ii) the diode polarity can be flipped by changing either the vortex sign, or the bias configuration. We argue that this may facilitate in-memory operation: an emerging new concept capable of boosting computer performance by avoiding bottlenecks associated with data shuffling between processor and memory[49]. This could open new perspectives for development of a digital superconducting computer. From this perspective it is advantageous that the diode is realized using conventional Nb-technology, which is mature enough for large-scale applications[6].

## Methods

**Samples.** The studied devices contain planar JJs. D1 is made from a Nb (70 nm, top)/CuNi(50 nm, bottom) bilayer with a superparamagnetic CuNi. D2 is made from a single Nb (70 nm) film. Films are deposited by dc-magnetron sputtering. They are first patterned into ~ 6 μm-wide bridges by photolithography and reactive ion etching, and subsequently nano-patterned by Ga$^+$ focused ion beam (FIB). Both Nb-CuNi-Nb (D1) and Nb-c-Nb (D2) JJs have a variable-thickness-bridge structure. They are made by cutting a narrow (20–30 nm) groove in the top Nb layer by FIB. Vortex trap (a hole ~ 50 nm in diameter) is also made by FIB. Both devices have a similar cross-like geometry, as can be seen in Fig. 3b and Supplementary Fig. 1. They have practically identical dimensions, specified in detail in the Supplementary. We fabricated and tested similar JJs with other metals in the bottom layer[3,42,43,45–48]. All of them work in a similar manner. Properties of Nb films are described in ref. [50].

**Experimental details.** Measurements were performed in a closed-cycle cryostat. Magnetic field is applied perpendicular to the film (positive $H$ along $y$-direction). $I_c(H)$ patterns were automatically recorded upon sweeping of magnetic field from a superconducting solenoid. For this current-voltage characteristics were examined and $I_c$ was determined using a small threshold voltage criterion, $V_{th} \sim 1 - 2$ μV. All the $I–V$s shown in the manuscript are nonhysteretic.

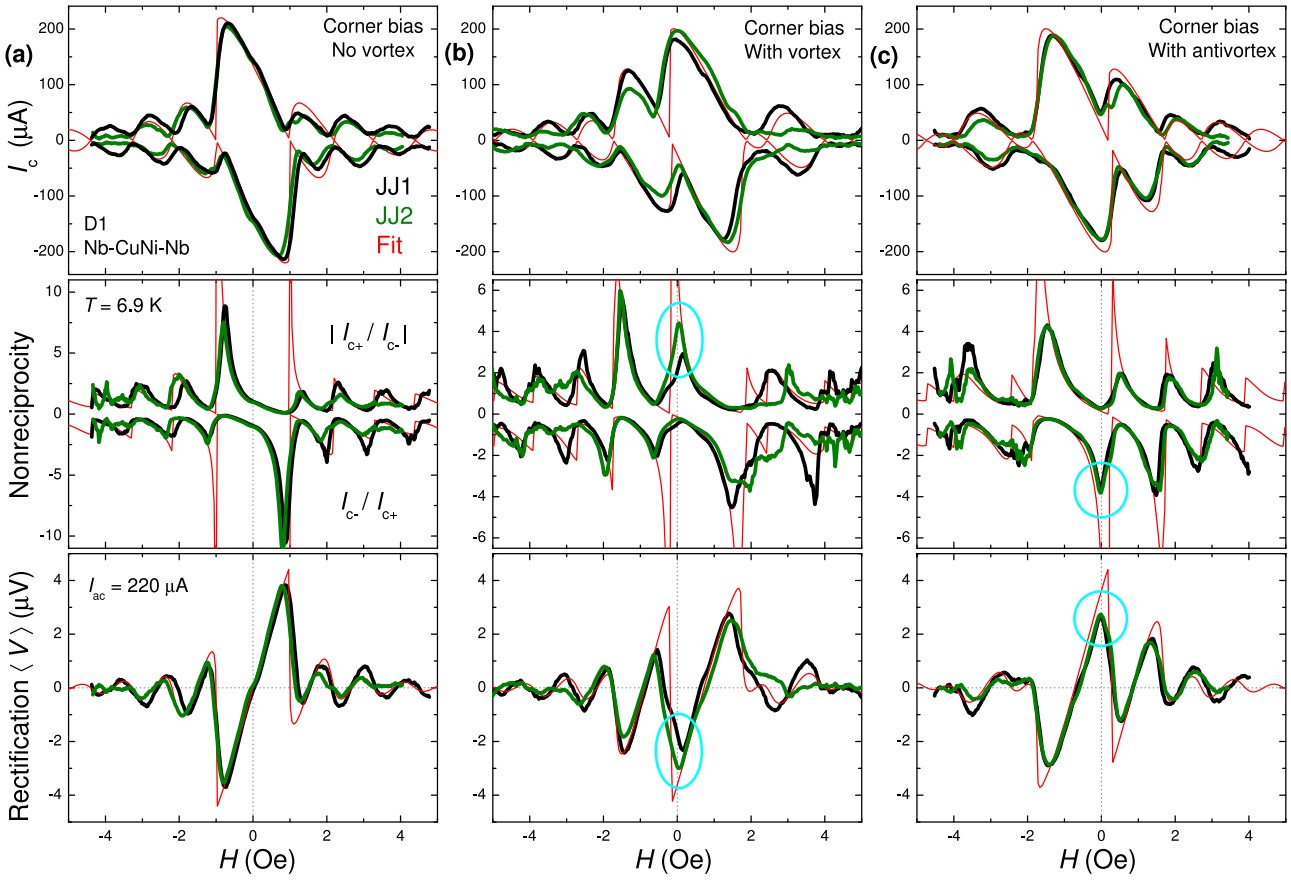

**Fig. 4 Diode operation on D1 with the right-corner bias. a** Without a vortex, **b** with a trapped vortex, and **c** with an antivortex. Top panels show the $I_c(H)$ modulations. Middle panels show the nonreciprocities, $|I_{c+}/I_{c-}|$, upper curves, and $I_{c-}/I_{c+}$, lower curves. Bottom panels show rectified dc-voltage calculated for harmonic ac-bias with $I_{ac} = 220\,\mu A \simeq I_{c0}$. Black and olive lines represent data for junctions 1 and 2 on D1, red lines are numerical fits. Appearance of a profound nonreciprocity and rectification at $H = 0$ is marked by cyan circles in **b**, **c**. All measurements are performed at $T \simeq 6.9$ K.

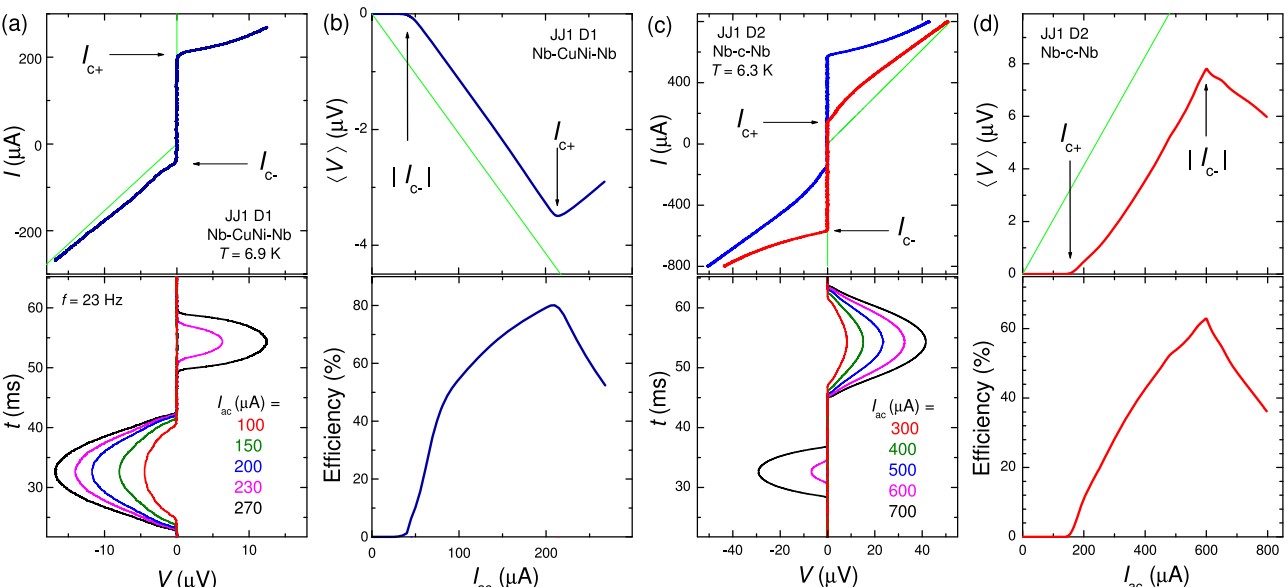

**Fig. 5 Amplitude dependence of rectification. a** Top: the $I–V$ of JJ1 on D1 (dark blue) with nearly maximum nonreciprocity, $I_{c+}/|I_{c-}|$. It is measured using right-corner bias, $I = I_{ac}\sin(2\pi ft)$. Green line represents the ideal case with infinite nonreciprocity, $I_{c-} = 0$. Bottom: Time dependencies of voltages during one ac-period at $f = 23$ Hz for different bias amplitudes. **b** Top: time-averaged dc-voltage as a function of ac-bias amplitude for JJ1 on D1 at $T \simeq 6.9$ K. Green line represents the ideal case, $\langle V \rangle_{max} = I_{ac}R_n/\pi$. Bottom: rectification efficiency, $\langle V \rangle/\langle V \rangle_{max}$, versus ac-bias amplitude. **c**, **d** These show similar data for JJ1 on D2 at $T = 6.3$ K. Top panel in **c** shows $I–V$s with maximum nonreciprocities obtained at $H = 0.7$ Oe (red) and -0.7 Oe (blue) without AV. **d** and bottom panel in **c** represent analysis of rectification for the red $I–V$ with $I_{c+} < |I_{c-}|$, leading to a positive rectified voltage.

**Numerical simulations**. The critical current is calculated by maximization of the Josephson current,

$$I_s = \int_0^L J_c(x) \sin[\varphi(x) + \varphi_0] dx, \tag{1}$$

with respect to the phase offset $\varphi_0$. Here $J_c(x)$ is the critical current density along the JJ. The Josephson phase difference $\varphi(x)$ has two contributions[41]:

$$\varphi(x) = \frac{2\pi d_{eff}}{\Phi_0} B_y x + \varphi_v(x), \tag{2}$$

where $d_{eff}$ is the magnetic thickness of the junction. Here the first term represents the linear phase gradient induced by the $y$-component of magnetic induction and the second—a nonuniform phase shift induced by the trapped Abrikosov vortex. Since we consider only short junctions, we neglect possible screening effects and assume that $B_y$ is uniform ($x$-independent). However, to account for the self-field effect we add an extra contribution to the applied external field $H$,

$$B_y = H + L_{sf} I_s / A, \tag{3}$$

proportional to the total current and the self-field inductance, $L_{sf}$. Here $A = L d_{eff}$ is the effective area of the junction. The vortex contribution is given by the azimuthal angle[43], indicated in the sketch in Fig. 1b,

$$\varphi_v(x) = -V \arctan\left(\frac{x - x_v}{|z_v|}\right), \tag{4}$$

where $V$ is the vorticity. Due to the self-field term in Eq. (3), $I_s$ is present in the right-hand-side of Eq. (1) as well. This implicit equation is solved iteratively using the bisection method. The fitting is obtained by varying two constants: $L_{sf}$ in Eq. (3) and $V$ in Eq. (4); as well as allowing for a nonuniform $J_c(x)$ distribution in Eq. (1). The fit represented by the magenta line in Fig. 1a and red lines in Fig. 4 corresponds to the V-shaped $J_c(x)$ with a 25% reduction in the middle of the JJ, $x = L/2$.

## Data availability

The data that support the findings of this study are available from the corresponding author upon reasonable request.

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

## Author contributions

T.G. fabricated samples and performed measurements with input from V.M.K. V.M.K. conceived the project and wrote the manuscript with input from T.G.

## Funding

## Competing interests

The authors declare no competing interests.
