## [Peer Review File · Nature Communications]

Demonstration of a superconducting diode-with-memory, operational at zero magnetic field with switchable non-reciprocityREVIEWER COMMENTS

Reviewer #1 (Remarks to the Author):

This work is a timely contribution on a rapidly advancing topic. The superconducting diode effect of 'Non-reciprocal superconducting-to-normal transition' attracts more and more attentions in the past one and a half years. This is most probably because a superconducting diode is promising for device applications with ultralow power dissipation, which is expected to significantly benefit future information technology. In previously reported SC diodes there are two major drawbacks that limit their real-world applications. One is the ultra-small ratio of the positive critical current to the negative critical current. The other is that previous SC diodes have to work in a magnetic field. In this manuscript, the authors overcome both limitations in a conventional Nb-based Josephson junction with a nonuniform current and a trapped flux quantum. They demonstrated a large nonreciprocity ratio of ~ 3 at zero magnetic field, presenting a significant achievement on superconducting diodes. This manuscript presents an original work based on well-established samples and experiments by the authors. The experimental results are clear and are well supported by numerical modellings. I recommend publication of this work on Nature Communications.

Below are some comments and/or confusing issues for the authors to consider to further improve this manuscript.

1. One critical concept of the newly studied SC diode effect is the one-way supercurrent. In this respect, I recommend the authors to show result of diode switching between zero-resistance state and normal-resistance state under zero magnetic field by alternating a positive and negative current. Since the diode nonreciprocity ratio of this work is high and the authors have well-established experimental techniques on this type of Josephson junctions, I guess this would not be a challenging task.

2. In order to clearly and directly demonstrate that the switching of the SC diode does be between the superconducting state and the 'normal' state, a comparison of the current dependent resistance under positive/negative currents ' $R(I+)$ vs $R(I-)$ ' is necessary. It would be better to show the results with both increasing and decreasing current(amplitude) sweeps to check the hysteresis effect of the Josephson junction. A neglectable hysteresis is favorable for real applications of SC diodes.

In addition, a temperature dependence of the junction resistance is useful for determining the value of the normal resistance as well as showing the T_c of the sample.

3. In Fig. 4b and 4c, the non-reciprocity and rectification peaks marked by green circles are not at $H = 0$ (It seems to be at a small positive field). Please clarify this.

The rectification results in 4b and 4c for the situations of with vortex and antivortex are shown under different ac current. Why to use different ac current? What's the ac current dependence of the rectification effect?

The authors obtained the rectification signal by averaging the time-dependent ac voltage (please correct me if my understanding is wrong). Could the authors show the time-dependent ac voltage? What's the frequency? What's the frequency dependence on the rectification? A discussion or estimation of the working frequency range would be very useful for guiding future applications of this SC diode.

4. Descriptions of fabrication and/or experiment methods are lacking. Although the authors stated 'Details of junction fabrication and characterization can be found elsewhere [3, 33, 34, 36, 37]'. In most or all of these references, there is a similar statement of 'details can be found elsewhere [xx]'. It is very challenging to trace the very first reference that contains the details. To improve the readability of this manuscript, I suggest the authors to directly add the corresponding detail information in the Method or in a Supplemental document for this manuscript. The parameters, such as the hybrid film thickness, the groove length, width and depth of the junction, the T_c of the sample, the temperature at which the experiments were conducted, the voltage criterion for extracting I_c , are all lacking.

5. What's the definition of length L ? The length of the junction groove? Could the authors mark it in Fig. 3a?

The x, y and z axis in Fig. 3a is misleading. I guess the authors mean that x and z axes are in the sample plane and y axis is out of plane. But the y axis does not look like to be out of plane. This is even confusing because usually x and y plane represents in-plane and z axis represents out of plane for the experiments in magnetic fields.

6. There are two other recent works Ref.31 and 'Zero-field superconducting diode effect in small-twist-angle trilayer graphene, Jiang-Xiazi Lin, et al. arXiv:2112.07841' on zero-field SC diode. Could the authors discuss the authors' scheme advantage over those works?

Reviewer #2 (Remarks to the Author):

The paper describes, with both numerical calculations and experiments, the use of trapped Abrikosov vortices in Josephson junctions to obtain a nonreciprocal supercurrent in zero applied field. While I believe that the paper is scientifically valid and its conclusions correct, I do not judge it sufficiently innovative for a Nature Communication publication (but I think it definitely deserves publication, for example in a more specialized journal) for the following reasons.

The physics studied in this manuscript is not completely new, a superconducting diode it has been studied by the very authors already long time ago (Ref [8], PRB 1997 - strictly speaking a supercurrent nonreciprocity can be obtained with an asymmetric SQUID as pointed out at least 40 years ago in the famous book of Barone and Paterno, Ch 12). What is presented as new here is the fact that the nonreciprocity is obtained at zero field.

The application of a vortex in a Josephson junction, and its impact on the effective current phase relation and junction characteristics, have already been discussed in other recent publications by the same authors, in particular Ref 34 (see section III and in particular the first paragraph for the References to previous works and their context). Also the very device has been already discussed elsewhere. Therefore, I judge the present contribution an incremental work based on previous studies by the authors, therefore more appropriate for a specialized journal.

The decision to publish this manuscript or not is, in the end, an editorial decision which shall be based on my comments (which I tried to expose here in the most transparent fashion) as well as on those of the other reviewers. It depends on the editorial standards of Nature Communication, about which the Editor has the final word. If the manuscript is nonetheless judged suitable for publication in Nature Communications, I would like the following comments/questions to be addressed.

1. The literature reported in the References about the superconducting diode effect is insufficient and misleading in the way it is presented, especially for what concerns the recent literature on spin-orbit induced diode effect. Under the name “superconducting diode” effect one finds in the recent literature two distinct effects:

(i) one is the nonreciprocal resistance, an effect that can be observed in normal noncentrosymmetric metal as pointed out by Rikken et al.[Phys. Rev. Lett. 87, 236602 (2001); see also review by Tokura and Nagaosa <https://doi.org/10.1038/s41467-018-05759-4>]. In superconductors, in the fluctuation regime

near T_c (where the resistance is still finite) the nonreciprocity is largely magnified. This effect is correctly represented by the References [23-25,27,28].

(ii) the second effect (see the seminal paper of Ando et al. Ref.[29]) is the spin-orbit induced nonreciprocal supercurrent in homogeneous superconductors. This is measured as either an asymmetry in the critical current [29,30] or in the Josephson inductance [30]. This effect it's all about the superfluid condensate, it does not rely on a finite resistance and it is therefore clearly distinct from (i). Many interesting papers appeared in the recent months about this subject, which is interesting because the nonreciprocal supercurrent becomes then an interesting probe of the physics of exotic superconductors, including magic angle twisted bilayer graphene. I list here a very minimal summary of these recent papers, which I would recommend to cite if the authors want to refer to the effect (ii).

-B. Pal, A. Chakraborty, P. K. Sivakumar, M. Davydova, A. K. Gopi, A. K. Pandeya, J. A. Krieger, Y. Zhang, M. Date, S. Ju, N. Yuan, N. B. M. Schröter, L. Fu, and S. S. P. Parkin, Josephson diode effect from Cooper pair momentum in a topological semimetal (2021), arXiv:2112.11285.

- J.-X. Lin, P. Siriviboon, H. D. Scammell, S. Liu, D. Rhodes, K. Watanabe, T. Taniguchi, J. Hone, M. S. Scheurer, and J. I. A. Li, Zero-field superconducting diode effect in twisted trilayer graphene (2021), arXiv:2112.07841.

- J. Diez-Merida, A. Diez-Carlon, S. Y. Yang, Y. M. Xie, X. J. Gao, K. Watanabe, T. Taniguchi, X. Lu, K. T. Law, and D. K. Efetov, Magnetic Josephson Junctions and Superconducting Diodes in Magic Angle Twisted Bilayer Graphene (2021), arXiv:2110.01067.

- L. Bauriedl, C. Bäuml, L. Fuchs, C. Baumgartner, N. Paulik, J. M. Bauer, K.-Q. Lin, J. M. Lupton, T. Taniguchi, K. Watanabe, C. Strunk, and N. Paradiso, Supercurrent diode effect and magnetochiral anisotropy in few-layer NbSe₂ nanowires (2021), arXiv:2110.15752.

- J. Shin, S. Son, J. Yun, G. Park, K. Zhang, Y. J. Shin, J.-G. Park, and D. Kim, Magnetic proximity-induced superconducting diode effect and infinite magnetoresistance in van der waals heterostructure (2021), arXiv:2111.05627.

Importantly, I recommend to change the final part of the second paragraph of the introduction.

>"Therefore, a zero field SC diode requires a specific violation of space-time symmetry. Recently it was suggested that this can be achieved with a help of non-centrosymmetric superconductors [23–25]"

[23-25] are not about zero-field SC diode, they discuss the effect (i) I mentioned above (where the field is required).

>"This renewed a search for a SC diode based on either exotic SCs [23, 25–28], or artificial heterostructures [29–31]."

This part is very confusing: it repeats Ref 23, 25, and refers to exotic SCs which are simply the noncentrosymmetric SCs just mentioned in the previous sentence. Importantly, it mixes then these works with those in Refs 29-31 which are about supercurrent diode effect (ii).

2. As mentioned in my general comment above, this paper is based on (many) other works by the same authors. For this reason, many experimental details are not described here: the authors refer instead to, e.g., Refs 34, 35, 36. This is legitimate, but in my view, this makes Section III difficult to read for the general audience of Nature Communications. I find that some key concepts should at least be at least briefly recalled. An example is Θ_v , which is introduced without clear definition and this makes the discussion in page 2 right column a bit obscure. The reader interested in technical details will definitely check the literature (e.g. Ref 34), but the reader interested in the essential physics should be able to obtain the main information in a self-contained way.

3. The temperature for the reported measurement is not mentioned. I would indicate also the T_c for the SC film (in the main text or in the Methods). I would also report (this perhaps in the supplementary) $R(T)$ curves. Other useful information will be the value of λ and λ_{Pearl} , and ξ , if available.

4. I find it useful for the reader if at least one panel of Fig1 contains a simple schematic of the geometry, containing the orientation of x,y,z axes and the corresponding orientation of H, J etc. This sketch could be also used to illustrate Θ_v , (see my comment 2 above) inspired e.g. by Fig.1 of Ref 34.

5. There is a lithographically-defined vortex trap in the middle of the junction, as seen in Fig3. But in the inset of panel (c) and (d) of Fig 3 there is a scheme where the vortex seems to be located out-of-center, in the corner (either left or right). This is confusing.

6. (minor point) I would not use the word “violating” in relation to symmetry: I would use “breaking” instead. I tend to associate the word violation with a law that is violated. In this case a symmetry is broken, since it refers to a boundary condition set by the experiment.

Reviewer #3 (Remarks to the Author):

The authors demonstrate a superconducting diode operation at zero magnetic field using a conventional Nb-based planar Josephson junction with a nonuniform bias and a trapped Abrikosov vortex. They also show that the critical current asymmetry $|I_{c+}/I_{c-}|$ can reach an order of magnitude and the rectification efficiency can exceed 70% with memory functionality. These achievements open a route for a new generation of superconducting in-memory computers. Thus, I support the publication of this paper in Nature Communications though the fundamental of the achievements is just a combination of known facts and techniques. I strongly recommend the authors to revise the manuscript for readability by considering the following points.

(1) In the section II, please add a figure to Fig.1 to illustrate the configuration of the numerical modelling. It was quite tiresome to imagine and confirm the device configuration by referring the SEM image in Fig.3(a). Important parameters such as x_v , z_v , should be indicated in the figure.

(2) In the Method section, the description of sample fabrication is not detailed enough.

What is the width of the bridge? What is the thickness of the bridge? It seems the authors prepared many devices with different thickness of the bridge. What is the difference in these devices?

There are many to be revised except the above points. Please read the manuscript as a reader who reads this manuscript for the first time.

Reply to Reviewer #1

Reviewer #1 writes:

”This work is a timely contribution on a rapidly advancing topic. The superconducting diode effect of ‘Non-reciprocal superconducting-to-normal transition’ attracts more and more attentions in the past one and a half years. This is most probably because a superconducting diode is promising for device applications with ultralow power dissipation, which is expected to significantly benefit future information technology. In previously reported SC diodes there are two major drawbacks that limit their real-world applications. One is the ultra-small ratio of the positive critical current to the negative critical current. The other is that previous SC diodes have to work in a magnetic field. In this manuscript, the authors overcome both limitations in a conventional Nb-based Josephson junction with a nonuniform current and a trapped flux quantum. They demonstrated a large nonreciprocity ratio of ~ 3 at zero magnetic field, presenting a significant achievement on superconducting diodes.

This manuscript presents an original work based on well-established samples and experiments by the authors. The experimental results are clear and are well supported by numerical modellings. I recommend publication of this work on Nature Communications. Below are some comments and/or confusing issues for the authors to consider to further improve this manuscript.”

Reply: We are grateful to the Reviewer for appreciation of our work and constructive critics and suggestions that helped to improve the manuscript. In the modified version we address all the issues raised by the Reviewer. We introduce significant changes (noted in red), add data for a new device (D2) and provide extended Supplementary information. Answers to specific questions are provided in replies below.

Reviewer #1 writes

”1. One critical concept of the newly studied SC diode effect is the one-way supercurrent. In this respect, I recommend the authors to show result of diode switching between zero-resistance state and normal-resistance state under zero magnetic field by alternating a positive and negative current. Since the diode nonreciprocity ratio of this work is high and the authors have well-established experimental techniques on this type of Josephson junctions, I guess this would not be a challenging task.”

Reply-1:

Thank you for the valuable suggestion. We provide additional clarifications about rectification of ac-signals in the new Fig. 5 and sec. IV of the Supplementary information. Supplementary Fig. 9 contains the required graphs. We also provide significant amount of additional information about rectification in the modified version.

Reviewer #1 writes

”2. In order to clearly and directly demonstrate that the switching of the SC diode does be between the superconducting state and the ‘normal’ state, a comparison of the current

dependent resistance under positive/negative currents ‘ $R(I+)$ vs $R(I-)$ ’ is necessary. It would be better to show the results with both increasing and decreasing current(amplitude) sweeps to check the hysteresis effect of the Josephson junction. A neglectable hysteresis is favorable for real applications of SC diodes.

In addition, a temperature dependence of the junction resistance is useful for determining the value of the normal resistance as well as showing the T_c of the sample.”

Reply-2:

In the Supplementary Figure 9 we demonstrate field and bias dependencies of dc-voltage for several positive and negative dc-currents for the cases without a vortex and with an antivortex. In the new Fig. 5 (b) and (d) we also explain dependencies of the rectified dc-voltage and the rectification efficiency on the ac-current amplitude. We also note that the maximum efficiency is close to 1-1/nonreciprocity.

The I-V curves of our diodes are non-hysteretic. All shown I-V’s contain data for both upward and downward bias sweeps. The corresponding statement is added in the manuscript.

Temperature dependencies are discussed in sec II of the Supplementary and Figures S3-S5.

Reviewer #1 writes

”3. In Fig. 4b and 4c, the non-reciprocity and rectification peaks marked by green circles are not at $H=0$ (It seems to be at a small positive field). Please clarify this.”

Reply 3.1:

To clarify this issue, we marked axes in Fig. 4. Now it is clearly seen that the positive peak in Fig. 4 (c) occurs exactly at $H=0$. The negative peak in Fig. 4 (b) is indeed slightly offset from zero by $\sim+0.1$ Oe. Ideally, patterns in Figs. 4 (c) and (d) should be centrosymmetric with respect to each other. Therefore, if the peak occurs at $H=0$ in (c), it should also be at $H=0$ in (b). However, in reality there is a slight non-centrosymmetric distortion, which might be caused by a small nonuniformity of the critical current density, $J_c(x)$, in the junction. In any case, even for the case of Fig. 4 (b) the offset is so small that the value of nonreciprocity at $H=0$ remains decent (more than two). In the modified version we added data for the second junction on the same device. Due to a slightly different distance from the trap, it appears to be better aligned for zero-field operation and demonstrates reciprocity of more than four at $H=0$ in the new Fig. 4 (b).

Reviewer #1 writes

“The rectification results in 4b and 4c for the situations of with vortex and antivortex are shown under different ac current. Why to use different ac current? What’s the ac current dependence of the rectification effect?”

Reply 3.2

Following the critics, we provide data at the same $I_{ac} = 220 \mu A$ in the modified Figs. 4. Such bias is nearly optimal for rectification. The dependence on the ac-amplitude is described in details in the new Fig. 5 and sec. IV of the Supplementary.

Reviewer #1 writes

“The authors obtained the rectification signal by averaging the time-dependent ac voltage (please correct me if my understanding is wrong). Could the authors show the time-dependent ac voltage? What's the frequency? What's the frequency dependence on the rectification? A discussion or estimation of the working frequency range would be very useful for guiding future applications of this SC diode.”

Reply 3.3

Time dependence is demonstrated in the new Fig. 5. Typically, we measure at a frequency $f = 23$ Hz. The frequency range in our setup is limited by RLC filtering to about a kHz. Fundamentally, the operation frequency range of a JJ is determined by the characteristic frequency, $f_c \sim I_c R_n / F_0$, which can be in excess of 10 GHz and up to THz range for our junctions. But, processing of so high frequencies requires a proper microwave design with coaxial cables to the sample and transmission lines at the chip. Such a technique is well established and is widely used for analysis of microwave properties of Josephson electronics. We added this clarification in the text and the Supplementary sec. IV.

Reviewer#1 writes:

“4. Descriptions of fabrication and/or experiment methods are lacking. Although the authors stated ‘Details of junction fabrication and characterization can be found elsewhere [3, 33, 34, 36, 37]’. In most or all of these references, there is a similar statement of ‘details can be found elsewhere [xx]’. It is very challenging to trace the very first reference that contains the details. To improve the readability of this manuscript, I suggest the authors to directly add the corresponding detail information in the Method or in a Supplemental document for this manuscript. The parameters, such as the hybrid film thickness, the groove length, width and depth of the junction, the T_c of the sample, the temperature at which the experiments were conducted, the voltage criterion for extracting I_c , are all lacking.”

Reply-4:

Following the recommendation, we provide more detailed description of sample fabrication in sec. I of the Supplementary. We explicitly show all the dimensions in Fig. S1 and subsequent text in the Supplementary. We also add specification of experimental conditions throughout the text.

Reviewer writes:

“5. What's the definition of length L? The length of the junction groove? Could the authors mark it in Fig. 3a?

The x, y and z axis in Fig. 3a is misleading. I guess the authors mean that x and z axes are in

the sample plane and y axis is out of plane. But the y axis does not look like to be out of plane. This is even confusing because usually x and y plane represents in-plane and z axis represents out of plane for the experiments in magnetic fields.”

Reply-5:

Following the critics, we added a sketch of geometry in Fig.1 (b) and modified Fig. 3 (a) to show “L” and to give a better perception of the y-axis direction. We also show this image with detailed specification of dimensions in the Supplementary Fig. 1. Concerning the choice of axes: to avoid confusion we follow the nomenclature from previous works. The line of reasoning is that for conventional junctions (xy) usually coincides with the junction plane and z-axis is going into the electrode. To avoid confusion with the y-axis direction we add a note that we operate with right-handed coordinates (in Fig.3 captions).

Reviewer writes:

“6. There are two other recent works Ref.31 and ‘Zero-field superconducting diode effect in small-twist-angle trilayer graphene, Jiang-Xiazi Lin, et al. arXiv:2112.07841’ on zero-field SC diode. Could the authors discuss the authors’ scheme advantage over those works?”

Reply-6:

Thank you for pointing out the relevant work. We add it to our list along with nine new references. Our devices have two main differences with diodes considered there. First is their simplicity: the operation principle is well understood and the fabrication procedure, based on conventional Nb-technology, is well established. This is advantageous for practical applications. Second is the tunability and switchability of our diode by bias current and vortex manipulation, respectively. This facilitates memory functionality, which is important for development of novel in-memory analog computation, discussed in the manuscript.

Reply to Reviewer #2

Reviewer #2 writes:

“The paper describes, with both numerical calculations and experiments, the use of trapped Abrikosov vortices in Josephson junctions to obtain a nonreciprocal supercurrent in zero applied field. While I believe that the paper is scientifically valid and its conclusions correct, I do not judge it sufficiently innovative for a Nature Communication publication (but I think it definitely deserves publication, for example in a more specialized journal) for the following reasons.

The physics studied in this manuscript is not completely new, a superconducting diode it has been studied by the very authors already long time ago (Ref [8], PRB 1997 - strictly speaking a supercurrent nonreciprocity can be obtained with an asymmetric SQUID as pointed out at least 40 years ago in the famous book of Barone and Paterno, Ch 12). What is presented as new here is the fact that the nonreciprocity is obtained at zero field.

The application of a vortex in a Josephson junction, and its impact on the effective current

phase relation and junction characteristics, have already been discussed in other recent publications by the same authors, in particular Ref 34 (see section III and in particular the first paragraph for the References to previous works and their context). Also the very device has been already discussed elsewhere. Therefore, I judge the present contribution an incremental work based on previous studies by the authors, therefore more appropriate for a specialized journal.”

Reply:

Indeed, physics behind our diodes is very simple and perfectly understood. The technology, based on conventional Nb, is also well established. However, we consider this not as a shortcoming, but rather as a major advantage for potential applications. Superconducting electronics needs a reliable, scalable, technically simple diode with high nonreciprocity at zero field. We demonstrate such, with an additional important feature: tunability and switchability, which can enable in-memory operation. To our opinion, the latter is at least equally important.

We fully agree that it is necessary to give a proper historical perspective. PRB-1997 contains the first demonstration of a JJ diode operation, such as rectification. But, nonreciprocal supercurrents in asymmetric SC devices (SQUIDs and JJs) were indeed known much earlier. Following the critics, we added a sentence in the Introduction with two new references, including the mentioned book by Barone and Paterno: “It is well known that nonreciprocity may appear in spatially asymmetric SC devices \cite{Likharev_1981,Barone_1982}. SC diodes, based on spatially nonuniform Josephson junctions (JJs), were demonstrated long time ago...”

Reviewer #2 writes:

“The decision to publish this manuscript or not is, in the end, an editorial decision which shall be based on my comments (which I tried to expose here in the most transparent fashion) as well as on those of the other reviewers. It depends on the editorial standards of Nature Communication, about which the Editor has the final word. If the manuscript is nonetheless judged suitable for publication in Nature Communications, I would like the following comments/questions to be addressed.

1. The literature reported in the References about the superconducting diode effect is insufficient and misleading in the way it is presented, especially for what concerns the recent literature on spin-orbit induced diode effect. Under the name “superconducting diode” effect one finds in the recent literature two distinct effects:

(i) one is the nonreciprocal resistance, an effect that can be observed in normal noncentrosymmetric metal as pointed out by Rikken et al.[Phys. Rev. Lett. 87, 236602 (2001); see also review by Tokura and Nagaosa <https://doi.org/10.1038/s41467-018-05759-4>]. In superconductors, in the fluctuation regime near T_c (where the resistance is still finite) the nonreciprocity is largely magnified. This effect is correctly represented by the References [23-25,27,28].

(ii) the second effect (see the seminal paper of Ando et al. Ref.[29]) is the spin-orbit induced

nonreciprocal supercurrent in homogeneous superconductors. This is measured as either an asymmetry in the critical current [29,30] or in the Josephson inductance [30]. This effect it's all about the superfluid condensate, it does not rely on a finite resistance and it is therefore clearly distinct from (i). Many interesting papers appeared in the recent months about this subject, which is interesting because the nonreciprocal supercurrent becomes then an interesting probe of the physics of exotic superconductors, including magic angle twisted bilayer graphene. I list here a very minimal summary of these recent papers, which I would recommend to cite if the authors want to refer to the effect (ii).

-B. Pal, A. Chakraborty, P. K. Sivakumar, M. Davydova, A. K. Gopi, A. K. Pandeya, J. A. Krieger, Y. Zhang, M. Date, S. Ju, N. Yuan, N. B. M. Schröter, L. Fu, and S. S. P. Parkin, Josephson diode effect from Cooper pair momentum in a topological semimetal (2021), arXiv:2112.11285.

- J.-X. Lin, P. Siriviboon, H. D. Scammell, S. Liu, D. Rhodes, K. Watanabe, T. Taniguchi, J. Hone, M. S. Scheurer, and J. I. A. Li, Zero-field superconducting diode effect in twisted trilayer graphene (2021), arXiv:2112.07841.

- J. Diez-Merida, A. Diez-Carlon, S. Y. Yang, Y. M. Xie, X. J. Gao, K. Watanabe, T. Taniguchi, X. Lu, K. T. Law, and D. K. Efetov, Magnetic Josephson Junctions and Superconducting Diodes in Magic Angle Twisted Bilayer Graphene (2021), arXiv:2110.01067.

- L. Bauriedl, C. Bäuml, L. Fuchs, C. Baumgartner, N. Paulik, J. M. Bauer, K.-Q. Lin, J. M. Lupton, T. Taniguchi, K. Watanabe, C. Strunk, and N. Paradiso, Supercurrent diode effect and magnetochiral anisotropy in few-layer NbSe₂ nanowires (2021), arXiv:2110.15752.

- J. Shin, S. Son, J. Yun, G. Park, K. Zhang, Y. J. Shin, J.-G. Park, and D. Kim, Magnetic proximity-induced superconducting diode effect and infinite magnetoresistance in van der waals heterostructure (2021), arXiv:2111.05627.

Importantly, I recommend to change the final part of the second paragraph of the introduction.

>"Therefore, a zero field SC diode requires a specific violation of space-time symmetry. Recently it was suggested that this can be achieved with a help of non-centrosymmetric superconductors [23–25]"

[23-25] are not about zero-field SC diode, they discuss the effect (i) I mentioned above (where the field is required).

>"This renewed a search for a SC diode based on either exotic SCs [23, 25–28], or artificial heterostructures [29–31]."

This part is very confusing: it repeats Ref 23, 25, and refers to exotic SCs which are simply the noncentrosymmetric SCs just mentioned in the previous sentence. Importantly, it mixes then these works with those in Refs 29-31 which are about supercurrent diode effect (ii). "

Reply-1:

Thank you for clarifications. We admit that our attempt to describe the whole field in one sentence was not successful. We follow the recommendation and provide more accurate description of the two mentioned SOI-induced diodes in the introduction, including recommended references. It is written in a new 3rd paragraph of the introduction:

"Recently it was shown that nonreciprocity can be induced in noncentrosymmetric SC by spin-orbit interaction (SOI) [25–28]. This renewed search for diode effects in

noncentrosymmetric SC [25, 28–30] and heterostructures [31–33]. SOI can induce asymmetry of either resistance in the fluctuation region near T_c [25–28, 30, 34], or supercurrent at low T [31, 32, 35–39]. However, SOI-based diodes require significant magnetic field. In several works zero-field SC diode operation was reported [33, 36], involving additional nontrivial effects. In this respect, nonreciprocity can be a tool for investigation of unconventional SC [34–39].”

Reviewer #2 writes:

“ 2. As mentioned in my general comment above, this paper is based on (many) other works by the same authors. For this reason, many experimental details are not described here: the authors refer instead to, e.g., Refs 34, 35, 36. This is legitimate, but in my view, this makes Section III difficult to read for the general audience of Nature Communications. I find that some key concepts should at least be at least briefly recalled. An example is Θ_v , which is introduced without clear definition and this makes the discussion in page 2 right column a bit obscure. The reader interested in technical details will definitely check the literature (e.g. Ref 34), but the reader interested in the essential physics should be able to obtain the main information in a self-contained way.”

Reply-2:

Thank you for the comment. In the modified version we expanded technical description to make the manuscript self-consistent. We also provide additional data for the second junction on the same device as well as data on a new sample D2, along with extended supplementary information.

Reviewer #2 writes:

“3. The temperature for the reported measurement is not mentioned. I would indicate also the T_c for the SC film (in the main text or in the Methods). I would also report (this perhaps in the supplementary) $R(T)$ curves. Other useful information will be the value of λ and λ_{Pearl} , and ξ , if available.”

Reply-3:

Following recommendations, we added corresponding information, including $R(T)$ to the Supplementary. In subsection “Parameters of Nb films” we describe characteristic lengths:

“Pearl length for similar Nb-CuNi-Nb JJs at $T=6.7$ K was estimated in Ref. [42] as, $\lambda_{\text{Pearl}} \approx 300$ nm. It was deduced from observation of a crossover to the mesoscopic limit for Abrikosov vortex-induced Josephson phase shift. Assuming the empirical temperature dependence of the London penetration depth $\lambda(T) = \lambda(0) [1 - (T/T_c)^4]^{-1/2}$ and taking $T_c=8.4$ K, this gives $\lambda(0)=112$ nm. The in-plane coherence length, $\xi(0) \approx 14$ nm, was estimated from analysis of the upper critical field in Ref. [S1].”

Reviewer #2 writes:

“4. I find it useful for the reader if at least one panel of Fig1 contains a simple schematic of the geometry, containing the orientation of x,y,z axes and the corresponding orientation of H, J etc. This sketch could be also used to illustrate Θ_v , (see my comment 2 above) inspired e.g. by Fig.1 of Ref 34.”

Reply-4:

We added a sketch in Fig. 1(b) and a Supplementary Fig.1 with specification of dimensions.

Reviewer #2 writes:

“5. There is a lithographically-defined vortex trap in the middle of the junction, as seen in Fig3. But in the inset of panel (c) and (d) of Fig 3 there is a scheme where the vortex seems to be located out-of-center, in the corner (either left or right). This is confusing.”

Reply-5:

There are no vortices in the data of Fig.3 (b-d). The circles with dot and cross were drawn to indicate directions of self-fields. We removed them, to avoid the confusion.

Reviewer #2 writes:

“6. (minor point) I would not use the word “violating” in relation to symmetry: I would use “breaking” instead. I tend to associate the word violation with a law that is violated. In this case a symmetry is broken, since it refers to a boundary condition set by the experiment.”

Reply-6: Done.

Final notes:

We agree that “The physics studied in this manuscript is not completely new”. But, it is not our goal to study interesting new physics. Our aim is applied: “Demonstration of a superconducting diode-with-memory, operational at zero magnetic field with switchable non-reciprocity”. The demonstrated prototypes are very simple and have superior characteristics, compared to earlier SC diodes. Experimentally achieved nonreciprocity is up to factor 10, numerically – 100, and theoretically – unlimited. We do not “hide” the simplicity, to the contrary we emphasize it (in the new version on p.2 we write: “Our concept has two SIMPLE ingredients: ...”). Since we are aiming at practical applications, simplicity and reliability, enabled by the well-developed Nb technology, are the key advantages. However, our work does contain significant technical innovations. First is the cross-like, four-terminal Josephson junction geometry. It allows tunable introduction of asymmetry and nonreciprocity. We can continuously tune the shape of $I_c(H)$ patterns (and IVs) from right-tilted, to straight and to left-tilted, as shown in Figs. 3(d,e). Such tunability is important for bringing the nonreciprocity to zero field and for changing the diode polarity. This is a unique feature of our device. Second is the vortex-assisted memory functionality. It could facilitate in-memory and quasi-analog (e.g. neuromorphic) computation, as discussed in the manuscript. We

believe that such components can revolutionize future superconducting computation, beyond the standard RSFQ CPU-centered architecture.

In the modified version we took into account all the critics and added significant amount of supplementary data, including measurements on a new improved and further simplified device based on a single Nb film. Finally, we want to thank the Reviewer for valuable critics that helped to improve the manuscript and hope that introduced changes and additional clarifications will help to bring forward applied significance of our work.

Reply to Reviewer #3

Reviewer #3 writes:

“The authors demonstrate a superconducting diode operation at zero magnetic field using a conventional Nb-based planar Josephson junction with a nonuniform bias and a trapped Abrikosov vortex. They also show that the critical current asymmetry $|I_{c+}/I_{c-}|$ can reach an order of magnitude and the rectification efficiency can exceed 70% with memory functionality. These achievements open a rout for a new generation of superconducting in-memory computers. Thus, I support the publication of this paper in Nature Communications though the fundamental of the achievements is just a combination of known facts and techniques. I strongly recommend the authors to revise the manuscript for readability by considering the following points.”

Reply:

Thank you for appreciation of our work and valuable suggestions. In the modified version we introduce corresponding changes, 9 new references and provide significant amount of supplementary information, including measurements on a new improved and further simplified devices based on a single Nb film.

Reviewer #3 writes:

“(1) In the section II, please add a figure to Fig.1 to illustrate the configuration of the numerical modelling. It was quite tiresome to imagine and confirm the device configuration by referring the SEM image in Fig.3(a). Important parameters such as x_v , z_v , should be indicated in the figure.”

Reply-1:

To clarify the geometry, we add a sketch in the inset of Fig. 1 (b). We also specified dimensions for the SEM image from Fig. 3 (a). However, since this tends to obscure the SEM image, we show it in the Supplementary information, Fig. S1.

Reviewer #3 writes:

“(2) In the Method section, the description of sample fabrication is not detailed enough. What is the width of the bridge? What is the thickness of the bridge? It seems the authors

prepared many devices with different thickness of the bridge. What is the difference in these devices?”

Reply-2:

In the modified version we add more clarifications about samples in the Supplementary.

Reviewer #3 writes:

There are many to be revised except the above points. Please read the manuscript as a reader who reads this manuscript for the first time.

Reply-3:

We made corresponding editorial changes to the text.

REVIEWERS' COMMENTS

Reviewer #1 (Remarks to the Author):

I would like to thank the authors for doing additional experiments and providing clarifications. The demonstration of a superconducting diode at ZERO magnetic field is one important achievement in this field, which is also the major goal for the recently published Nature paper [33] during the review process of this manuscript. This manuscript realized the field-free diode by using an established technique and the well understood theory, therefore the method demonstrated in this manuscript could be more favored for future application of SC diodes.

Reviewer #2 (Remarks to the Author):

The authors have addressed the remarks of the Referees in a satisfactory fashion.

Concerning the innovations introduced in this work, my point of view is similar to that of Referee 3, namely, this work assembles known results obtained by the very same authors elsewhere.

I, however, agree with the authors that an innovative paper must not necessarily show new physics, and that new applications are interesting as well.

The new version of the manuscript is now clear, well written, the experimental evidences support the model proposed by the authors.

I therefore support publication of the new version of the manuscript.

Reviewer #3 (Remarks to the Author):

In my view the authors have responded satisfactorily to the questions raised by all three reviewers.

Hence, I support publication in Nature Communications.

Reply to REVIEWERS' COMMENTS

Reviewer #1 (Remarks to the Author):

I would like to thank the authors for doing additional experiments and providing clarifications. The demonstration of a superconducting diode at ZERO magnetic field is one important achievement in this field, which is also the major goal for the recently published Nature paper [33] during the review process of this manuscript. This manuscript realized the field-free diode by using an established technique and the well understood theory, therefore the method demonstrated in this manuscript could be more favored for future application of SC diodes.

Reviewer #2 (Remarks to the Author):

The authors have addressed the remarks of the Referees in a satisfactory fashion. Concerning the innovations introduced in this work, my point of view is similar to that of Referee 3, namely, this work assembles known results obtained by the very same authors elsewhere. I, however, agree with the authors that an innovative paper must not necessarily show new physics, and that new applications are interesting as well. The new version of the manuscript is now clear, well written, the experimental evidences support the model proposed by the authors.

I therefore support publication of the new version of the manuscript.

Reviewer #3 (Remarks to the Author):

In my view the authors have responded satisfactorily to the questions raised by all three reviewers. Hence, I support publication in Nature Communications.

Reply:

We are grateful to all three reviewers for appreciation of our work. Since the Reviewers were satisfied by the 2nd revision, in the final version we made only small editorial polishing without any significant changes. We added two new references: one is a newly published relevant work [28] and the other [50] is moved from the supplementary.